# High Predictive Value of the Soluble ZEBRA Antigen (Epstein-Barr Virus Trans-Activator Zta) in Transplant Patients with PTLD

**DOI:** 10.3390/pathogens11080928

**Published:** 2022-08-17

**Authors:** Julien Lupo, Anne-Sophie Wielandts, Marlyse Buisson, CRYOSTEM Consortium, Mohammed Habib, Marwan Hamoudi, Patrice Morand, Frans Verduyn-Lunel, Sophie Caillard, Emmanuel Drouet

**Affiliations:** 1Institut de Biologie Structurale, Université Grenoble-Alpes, 38000 Grenoble, France; 2Laboratoire de Virologie, Institut de Biologie-Pathologie, Centre Hospitalier Universitaire Grenoble Alpes, 38000 Grenoble, France; 3CRYOSTEM Consortium: Marseille Innovation—Hôtel Technologique, 13382 Marseille, France; 4Department of Microbiology, UMC Utrecht, 3584 CX Utrecht, The Netherlands; 5Département de Néphrologie et de Transplantation Centre, Hospitalier Universitaire de Strasbourg, 67091 Strasbourg, France

**Keywords:** Epstein-Barr Virus, ZEBRA protein, BZLF1, PTLD, post-transplant lymphoproliferative disease, predictive biomarker

## Abstract

The ZEBRA (Z EBV replication activator) protein is the major transcription factor of EBV, expressed upon EBV lytic cycle activation. An increasing body of studies have highlighted the critical role of EBV lytic infection as a risk factor for lymphoproliferative disorders, such as post-transplant lymphoproliferative disease (PTLD). We studied 108 transplanted patients (17 PTLD and 91 controls), retrospectively selected from different hospitals in France and in the Netherlands. The majority of PTLD were EBV-positive diffuse large B-cell lymphomas, five patients experienced atypical PTLD forms (EBV-negative lymphomas, Hodgkin’s lymphomas, and T-cell lymphomas). Fourteen patients among the seventeen who developed a pathologically confirmed PTLD were sZEBRA positive (soluble ZEBRA, plasma level above 20 ng/mL, measured by an ELISA test). The specificity and positive predictive value (PPV) of the sZEBRA detection in plasma were 98% and 85%, respectively. Considering a positivity threshold of 20 ng/mL, the sensitivity of the sZEBRA was 82.35% and the specificity was 94.51%. The mean of the sZEBRA values in the PTLD cases were significantly higher than in the controls (*p* < 0.0001). The relevance of the lytic cycle and, particularly, the role of ZEBRA in lymphomagenesis is a new paradigm pertaining to the prevention and treatment strategies for PTLD. Given the high-specificity and the predictive values of this test, it now appears relevant to investigate the lytic EBV infection in transplanted patients as a prognostic biomarker.

## 1. Introduction

The Epstein-Barr virus (EBV) infects more than 90% of the world’s population and remains in the system of infected individuals throughout their lifetime, most often without any severe or life-threatening consequences, because it is well controlled by our immune system. EBV is one of the most common human viruses and is the cause of pathologies such as infectious mononucleosis (IM) and certain cancers—such as immunodeficiency-related B cell lymphomas, Burkitt and Hodgkin lymphomas, nasopharyngeal, and gastric carcinomas [1]. Recent findings have shown that there is a strong association between multiple sclerosis and EBV [2]. EBV has the ability to switch between two alternative phases: latent or lytic replication. EBV lytic replication, which is required for the horizontal spread of the virus from one cell to another, and from host to host, infects both the epithelial cells and the B cells [3]. EBV is a herpesvirus (type 4), with a genome encoding 86 proteins [4]. The core set of genes (a minority) are involved during the latency phase. The other set comprises the genes of the lytic cycle. For many years, researchers claimed that only the products of the latency genes (the LMP-1 oncoprotein and the EBNAs) were responsible for oncogenesis. It has recently been demonstrated that the proteins of the lytic cycle have a role, not only in cell transformation (the initial stage of the tumor process), but also in tumor progression [5,6,7,8]. In transplant patients, an activation of EBV replication and an absence of immune control of the proliferation of infected B cells may lead to the emergence of a post-transplantation lymphoproliferative disorder (PTLD). These PTLDs are classically described as malignant proliferations of the lymphoid type, but they group together a wide variety of histological and immunological types [9], including several types of lymphomas (B-cell lymphomas—most often linked to EBV—but also T-cell lymphomas, Burkitt’s lymphomas, and Hodgkin’s disease). The diagnosis and treatment of PTLD remains difficult, with a mortality rate close to 30% [10,11]. EBV is present in approximately 80% of these PTLDs. Indeed, recent studies have revealed a seamlessness between latent and lytic proteins and the types of infections to which they contribute [12,13]. Some lytic proteins, such as the immediate early (IE) ZEBRA (Z EBV replication activator or Zta) protein, can be expressed in the context of latent infections, as seen in some cancers and in the pre-latent infection of B lymphocytes [6]. Several experiments highlighted the role of IE lytic viral protein expression in the lymphomagenesis of immunocompromised mice [14]. Other studies pointed out the crucial role of lytic EBV infection in the development of B-cell lymphomas in thymic tissue-reconstituted mice or cord blood-humanized mice [15,16]. The BZLF-1 encoded ZEBRA protein is the major transcription factor of EBV, required for the activation of the EBV lytic cycle, while directly regulating the expression of a cellular gene set [17,18]. Overall, these studies have shown that the presence of a limited number of cells lytically infected with EBV may enhance tumor growth through the release of growth factors and immunosuppressive cytokines [6].

Usually, the diagnosis of an EBV primary infection is established by a serodiagnosis (the detection of antibodies specific to EBV) [19]. In the context of a nasopharyngeal carcinoma (NPC), this serology can also be applied to consolidate a diagnosis evaluation or follow a therapy [20]. By contrast, in the case of lymphomas, clinicians have chosen molecular biology techniques (viral load by the quantification of EBV DNA circulating in the blood) [21]. This technique, measuring the viral genome copy number in latently infected memory B cells and in the plasma extracellular compartment, is routinely used to monitor transplant patients. However, this method has some limitations, and not all patients at risk of PTLD can be identified by EBV DNA measurements alone. This is particularly important when there is a chronic long-term carriage of a high EBV load in young pediatric solid organ transplant (SOT) patients [22,23,24,25,26]. Moreover, this technique lacks sensitivity in EBV-negative lymphomas [27]. Below, we describe a new ELISA-based serological biomarker, detecting the soluble ZEBRA protein for the detection of the lytic cycle in transplant patients. This first-in-class test was developed in order to improve the specificity of the diagnosis of post-transplant lymphomas.

## 2. Results

### 2.1. Patients

Among the 17 PTLD cases, 55% were women, compared to 41% in the control patients (*p* = 0.5). The median age of the cases and controls at the time of transplant was similar (57 years). There was no difference in the distribution of the type of organ transplanted between the cases and the controls (*p* = 0.2). Regarding EBV pre-transplant serology, more D/R mismatches were observed in the cases: four out of nine patients (2 D+/R−, 2 D−/R+), compared to 5 out of 50 mismatches in the control population (1 D+/R−, 4 D−/R+). We did not observe any difference between the cases and controls regarding the use of T-cell depleting agents or the number of HLA mismatches between the donor and the recipient. There was no significant difference in the use of anti-CMV prophylaxis between the cases and controls (*p* = 0.2). Regarding anti-rejection maintenance therapy, the most commonly used calcineurin inhibitor in the cases was ciclosporin (81%), while in the controls it was more often tacrolimus (95%). HSCs transplant recipients developed PTLD earlier than lung and kidney transplant recipients, after a median of 4 months, 19 months, and 8 years, respectively (Table 1 and Table 2).

### 2.2. Characterization of EBV Markers in Patients with or without PTLD

#### 2.2.1. Quantification of EBV Viral Load and Detection of sZEBRA (Soluble ZEBRA)

The EBV DNA load in whole blood was not different between the patients with PTLD and those without PTLD. We established the cut-off point for the sZEBRA test based on the mean OD_450_ obtained from the seronegative EBV sera present in each ELISA experiment. The threshold was set at 0.211, corresponding to the mean OD_450_ obtained over 30 experiments (0.115), plus two standard deviations (0.048). The mean OD_450_ (+/− standard error of the mean) of sZEBRA was significantly higher in patients with PTLD than in transplant patients without PTLD (*p* < 0.0001) (Table 3 and Figure 1). Among the six sZEBRA-positive cases, only one patient had a detectable viral load. There was no association between a detectable EBV viral load and sZEBRA detection. Among the five sZEBRA-positive control patients, two had a detectable EBV DNA load.

#### 2.2.2. Diagnostic Value of sZEBRA Antigen

We constructed a ROC curve to obtain the diagnostic values of the ELISA, based on 17 samples from the patients who developed PTLD, and 91 samples from the control patients (Figure 2). Considering a positivity threshold of OD_450_ at 0.211 (mean + 2SD), the sensitivity was 82.35% (95% CI 58.97, 93.81) and the specificity was 94.51% (95% CI 87.78, 97.63). Considering the prevalence of PTLD in the study population (16%), the positive predictive value (PPV) was 74%. It was 14.2% when considering a PTLD prevalence of 1% (corresponding to that of the general transplant population). The negative predictive values were 97% or 99.8%, respectively. If the threshold was set at OD_450_ at 0.259 (mean + 3SD), the sensitivity was 64.71% (95% CI 41.30, 82.69) and the specificity was 97.80% (95% CI 92.34–99.61). The positive predictive value was 85% when taking the prevalence at 16%, or 24.8% when considering a prevalence of 1%. The negative predictive values were 94% and 99.6%, respectively (Table 4).

## 3. Discussion

In 2017, it was estimated that approximately 100,000 people worldwide benefitted from a hematopoietic stem cell transplant (allograft) [28], including 42,000 in Europe [29]. In addition, the Health Resources and Services Administration (HRSA) recorded over 124,000 SOT cases (+15% in 5 years). Nevertheless, immunosuppressive treatments essential to the success rate of these transplants, lead to a risk in the appearance of potentially fatal malignant lymphoproliferations, grouped under the term PTLD. The diagnosis and treatment of PTLD remain difficult, with a mortality rate close to 30% [10,11]. Approximately 80% of these PTLDs are associated with EBV (i.e., the presence of the virus in tumors).

The occurrence of EBV-associated PTLDs is generally preceded by an increase in a reactivation-related EBV DNA load and an increase in the number of infected B cells [21]. Regular PCR monitoring of the EBV DNA load in the whole blood or plasma of transplant recipients is, therefore, currently recommended to allow for pre-emptive therapy, based on immunosuppressive modulation and the use of a monoclonal anti-CD20 antibody targeting B lymphocytes (Rituximab) [11,30]. A variety of methods, gene targets, and blood compartments (leukocytes, whole blood, plasma, and serum) are being used. Interlaboratory variability in the results is considerable (100–10,000 fold differences), making comparisons across institutions difficult [31]. Prospective studies, especially in adult seropositive recipients, correlating the specific viral loads to the risk of PTLD have been inconsistent. For now, no clear guidelines exist [32]. Despite this, EBV DNAemia detection and quantification by molecular assays during the post-transplant period is the gold standard method to diagnose infection, guide preemptive strategies, and monitor the response to therapy [33,34]. As a result, the optimal blood compartment to test and the optimal viral load cut-off to use for initiating and interrupting preemptive therapy is unclear.

Moreover, the EBV load in the whole blood of adults with PTLD after SOT does not correlate with the clinical course [25], which could lead to the late diagnosis of PTLD or to unnecessary investigations. It is worth mentioning that rituximab, a monoclonal antibody medication used in the treatment of lymphoproliferative disorders, is a drug with adverse side effects, especially in SOT recipients, as they require much higher doses. The doses are sometimes so high that the effects of hypogammaglobulinemia can lead to chronic and/or severe superinfections, which are often debilitating [35,36]. Other virological or immunological biomarkers are, therefore, necessary to complete the measurement of the EBV blood viral load and to improve its predictive value. Several serological and anatomo-pathological arguments suggest that the viral protein, ZEBRA, may be a candidate marker for EBV-associated lymphomas, particularly for the post-transplantation lymphoproliferative syndromes [37,38,39,40]. In a recent review, it was proposed that ZEBRA could be involved in EBV pathogenesis, not only as an essential protein for EBV replication activation but also as a “toxin” released into the extracellular milieu (Appendix A) [6]. All in all, a likely hypothesis is that early abortive infections, associated with fully lytic cycles, may occur in the tumor or its environment, eventually releasing ZEBRA into the bloodstream. For the first time, it was possible to detect the soluble ZEBRA (sZEBRA) protein in the serum/plasma of transplant recipients (measured by an antibody-based ELISA) [41]. These positive results proved the clinical benefits of assaying the EBV ZEBRA viral protein in serum or plasma from patients at the time of developing post-transplant syndrome.

Our study population included 17 patients with PTLD, selected retrospectively from French and Dutch cohorts and 91 control patients, transplanted without PTLD, taken from the Grenoble trials. Several studies indicated that the risk of developing lymphoproliferative syndrome depended on the type of organ transplanted, the occurrence of GvHD, the EBV and HLA system mismatches between the donor and the recipient, and the types of immunosuppressants administered [9,42]. As described in the attached documentation, this study showed that it is possible to identify such risk factors, namely EBV serology mismatches between the donor and the recipient. It was also observed that HSCs transplant recipients developed PTLD earlier than lung and kidney transplant recipients. Related information on PTLD can be found in the attached documentation [43]. Regarding immunosuppressive therapy, the greater use of tacrolimus in the controls than in the cases may be related to the more recent inclusion period of the controls compared to the cases [44]. Concerning the EBV viral load, it is stated that patients with PTLD have an EBV viral load that is more easily detected and is higher than in patients without diseases associated with EBV [9,32].

The results of the study did not show a significant difference in the EBV DNA amount between the patients with PTLD and the controls without PTLD. Although decreasing in number due to pre-emptive therapy strategies based on the monitoring of the EBV viral load in blood by PCR, EBV-related PTLD cases remain a serious and potentially fatal complication. The presence of the soluble ZEBRA antigen, which is soluble in the serum of transplant patients [41], makes it possible to characterize the level of the pathogenic reactivation of the EBV lytic infection more accurately, which is, in itself, a risk factor for lymphomagenesis in animal models [15]. The study showed that the new antigenic marker, sZEBRA, was found more frequently and at higher values in the PTLD patients than in the control patients. This confirms the results obtained in the study by Habib et al. [41]. In this last study, the viral load in whole blood was not associated with the presence of sZEBRA in plasma. This could be due to the different matrix used for these two analyses (PCR and ZEBRA), as no correlation could be found between the viral load in the cell compartment and the plasma, as already described [45]. Despite a negative EBV DNAemia in four patients who developed PTLD (cases of EBV-negative tumors) (Table 2), sZEBRA antigen has been detected. A hypothesis was raised that abortive lytic cycles could take place in the tumor or its environment, as demonstrated elsewhere [12,41]. EBV may also participate in the early development of the tumor and then disappear (“hit-and-run”) [9]. The study was retrospective, but some limitations need to be addressed. First, the number of patients with PTLD was low, and this fact limited the strength of the statistical investigations. Second, the patient samples were collected retrospectively, and were initially obtained exclusively for routine clinical purposes. The sampling schedule of some specimens in relation to PTLD diagnosis was, therefore, not accurately performed. The data suggested that sZEBRA detection in transplant patients could be a risk factor for PTLD. Patients with sZEBRA levels >20 ng/mL should be given the highest priority for close clinical and biological monitoring and receive preemptive treatment if there is evidence of sequential rising levels. Further research is needed to prospectively evaluate the risk of PTLDs with this novel biomarker.

This study clearly demonstrates the following: (i) either the lytic infection itself or the expression of ZEBRA or ZEBRA-controlled genes play a key role in the development of EBV-induced cancers, and (ii) sZEBRA detection is definitely a first-in-class early biomarker, to be used to follow-up patients who are at risk. Furthermore, this could eventually raise the possibility that the lytic antigens (including ZEBRA) might be useful therapeutic or vaccine targets for the prevention of EBV-induced cancers [46].

## 4. Materials and Methods

### 4.1. Patients

The recruitment of transplant patients (Solid organ and hematopoietic stem cells transplants) with a PTLD episode was retrospective and multicentered (COLT/CHU Nantes cohort, Strasbourg University Hospital cohort, CRYOSTEM Consortium (https://openbioresources.metajnl.com/articles/10.5334/ojb.58/ (accessed on 3 April 2020) and the SFGM-TC, UMC Utrecht cohort). The clinical characteristics of the populations (cases and controls) included in the study are shown in Table 5, and those of all the patients initially enrolled in the study are in Table 1 and Appendix A. The cases of PTLD had occurred between 1986 and 2016. Because of this, for several cases, information such as donor/recipient EBV status or tumor EBV status could not be obtained. The recruitment of the control population (patients transplanted without PTLD) was carried out prospectively at the Grenoble University Hospital, between April and September 2019. The samples collected for the study were whole blood, serum, and plasma samples. For each patient, the date of birth, the nature and date of the transplant, the serological status EBV D/R before transplant, immunosuppressive therapy, and EBV viral load were obtained. If the information was available, the other data recorded were the use of anti-CMV prophylaxis and the risk factors for developing PTLD, i.e., HLA pairing, T cell depletion, and status and date of a possible GvHD. PTLD diagnosis was based on examining histological material obtained by either open biopsy or core needle biopsy, with lesions classified according to the WHO classification of tumors [47,48]. Association with EBV was confirmed by in situ staining for EBER (Epstein-Barr encoding region).

### 4.2. Quantification of EBV DNA in Whole Blood

The laboratory performed EBV DNA load in the control patients (cases was collected from the participating centers). EBV viral load was measured by qPCR in whole blood. DNA was extracted from 200 μL of whole blood with the NucliSens EasyMag (BioMérieux, Marcy l’Etoile, France). Amplification was done using the commercial EBV R quantification kit gene (BioMérieux, France) on the LightCycler 480 platform (Roche Diagnostics, Meylan, France). The viral load was expressed in copies/mL. The limit of detection was 200 copies/mL, and the limit of quantification was 500 copies/mL. In order to compare viral loads between cases (values expressed in UI/mL) and controls, we homogenized the expression of the results in IU/mL, by applying a conversion factor of 2.09 to the viral load result of controls expressed in copies/mL [49].

### 4.3. Detection of Soluble ZEBRA (sZEBRA) Protein in Plasma Samples

The ZEBRA antigen assay is a monoclonal antibody (Mab)-based “sandwich” ELISA test. The anti-ZEBRA monoclonal antibodies used in the assay were AZ69 capture antibody targeting the N-ter part, and AZ130 revealing antibody targeting the C-ter part (patent PCT/FR2012/052790). The calibration ZEBRA protein was produced in CHO cells by the company Biotem, Apprieu, France. The saturation and dilution buffer consisted of 4% gelofusin, 10% FCS, 1% Tween 20, and 1X PBS, and the wash buffer of 0.05% Tween 20 and 1X PBS. All the plasma samples were first decomplemented (30 min at 56 °C) before analysis. The protocol of the test was described elsewhere [41]. Briefly, the AZ69 capture antibody was coated in a 96-well plate overnight, at +4 °C. After washing, the plates are saturated in PBS-gelatin buffer. Plasmas diluted to 1/10 or the antigen ZEBRA (ranged from 400 to 2.5 ng/mL) were incubated at room temperature for 1 h. Next, the samples are incubated with AZ130 revealing antibody, coupled to biotin. After washing, streptavidin-HRP was added to the wells for 30 min in the dark. After 3 washes, the revelation was carried out by adding 100 μL of TMB in each well. After 10 min of incubation of the plate at room temperature, with stirring in the dark, the reaction was stopped by adding 100 μL of 0.1 M H2SO4. The plate was read at 450 nm and 630 nm (reference), using a Biotek EL 808 spectrophotometer (Biotek Instruments, Inc., Winooski, VT, USA) and using Gen5 software. The quantitative assay was performed using a sZEBRA concentration calibration curve.

### 4.4. Statistical Analysis

The exact Fisher test was performed to compare categorical variables (distribution men-women, risk factors, anti-CMV prophylaxis, and EBV DNA load). For the variables of age and the time of onset of PTLD by organ, we presented the results as median and interquartile range. The values of sZEBRA (OD_450_) in cases and controls were compared using the Mann–Whitney nonparametric test. Regarding value diagnostic of the ZEBRA antigen, the ROC curve was calculated with the 95% confidence interval by the Wilson–Brown method. For each test, a *p*-value with a two-tailed test <0.05 was considered statistically significant. Statistical analysis was performed using Graphpad 8.4.3 software.

### 4.5. Ethical Considerations

This non-interventional retrospective study, involving data and samples from human participants was carried out in Grenoble University Hospital, according to French current regulation. The investigators have signed a commitment to comply with Reference Methodology n°004, issued by French Authorities (CNIL). Subjects were all informed and did not oppose. Written consent was not required for the retrospective study, in accordance with the national legislation and the institutional requirements. For the prospective inclusions, written consent was obtained from all patients to participate in the virology collection DC-2008-680. The raw data supporting the conclusions of this article will be made available by the authors with respect of the General Data Protection Regulation, without undue reservation.

## 5. Conclusions

The relevance of the lytic cycle and particularly the role of ZEBRA in lymphomagenesis is a new paradigm pertaining to the prevention and treatment strategies for PTLD. Given the good specificity and positive predictive value of this test, it now appears relevant to investigate the lytic EBV infection in transplanted patients as a prognostic biomarker.

## Figures and Tables

**Figure 1 pathogens-11-00928-f001:**
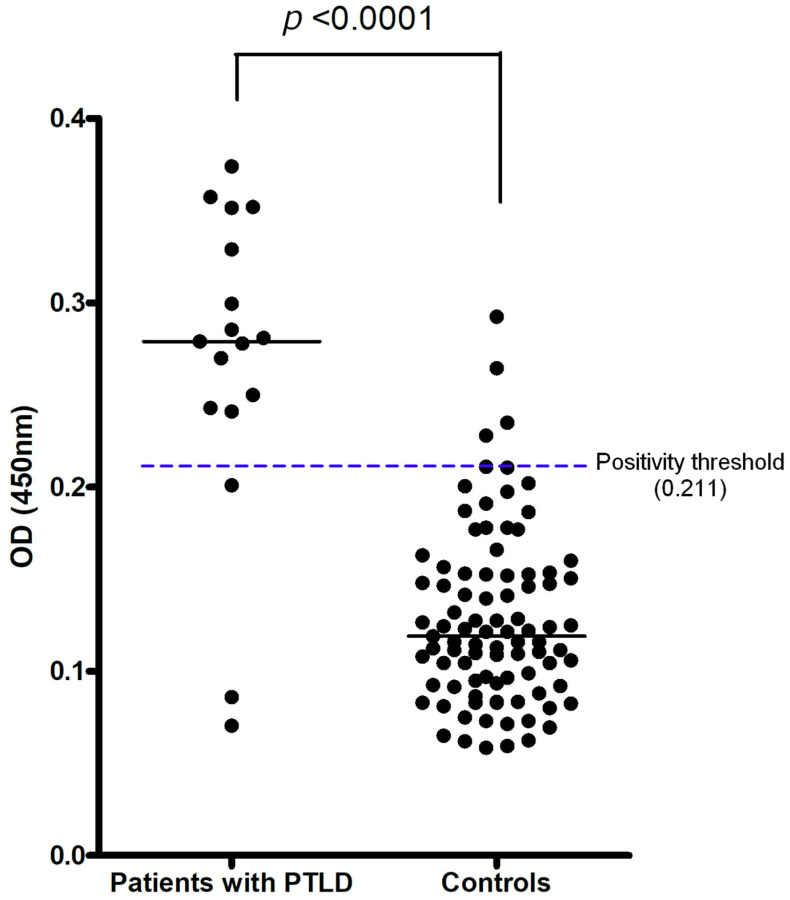
Distribution of OD_450_ in the ELISA assay for detection of sZEBRA (soluble ZEBRA), in cases (PTLD) and controls (transplanted without PTLD). The mean OD_450_ (+/− standard error of the mean) of sZEBRA was significantly higher in patients with PTLD than in transplant patients without PTLD (*p* < 0.0001). The positivity threshold was defined as the mean + 2SD (standard deviation) of 30 seronegative plasmas. A patient is considered positive (+) for sZEBRA (soluble ZEBRA) if the ELISA OD_450_ on a serum sample is greater than 0.211 (mean + 2SD of 30 seronegative plasmas, corresponding to 20 ng/mL). The calibration curve to obtain the direct correlation between OD_450_ and sZEBRA concentration is provided in the Appendix A.

**Figure 2 pathogens-11-00928-f002:**
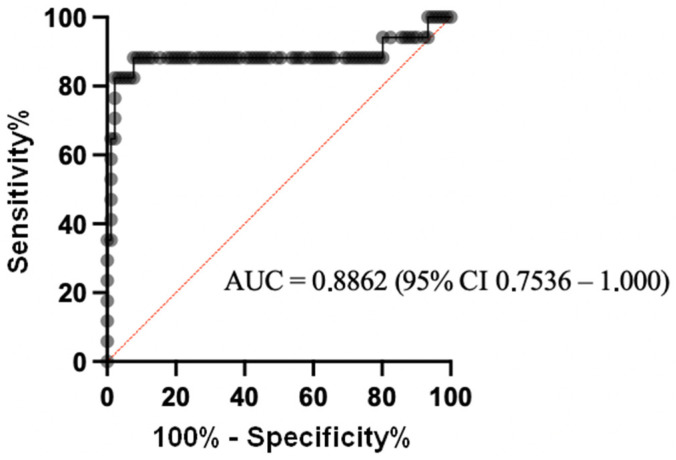
ROC curve of the sZEBRA assay by ELISA on decomplemented plasma samples. The diagnostic values of this test were based on 17 samples from patients who developed PTLD and 91 samples from control patients.

**Table 1 pathogens-11-00928-t001:** Clinical characteristics of the case and control patients. The most frequently used anti-rejection maintenance therapies in cases and controls were calcineurin inhibitor and tacrolimus, respectively. HSCs transplant recipients developed PTLD earlier than lung and kidney transplant recipients. F—female; M—male; HSCs—hematopoietic stem cells; D/R—donor/recipient; IS—immunosuppressants; PTLD—post-transplantation lymphoproliferative disease; DLBCL—diffuse large B-cell lymphoma; IM-like—infectious mononucleosis like.

		Cases (17 Patients)		Controls (91 Patients)		*p*-Value
**F** n (%)	**M** n (%)	6/11 (55)	5/11 (45)	37/91 (41)	54/91 (59)	0.5
**Age Q2 (Q1–Q3)**		57 (27.5–64.5)		57 (40–65)		0.8
**Organ** n (%)						0.2
Kidney		7 (41%)		20 (22%)		0.1
Lung		3 (18%)		33 (36%)		0.2
HSCs		7 (41%)		38 (42%)		1
**Risk Factor** n (%)						
Mismatch EBV D/R		4/9 (44%)		5/50(10%)		0.02
T-cell depletion		3/8 (38%)		12/29 (41%)		1
Mismatch HLA D/R		9/12 (75%)		28/33 (85%)		0.7
**Anti-CMV** n (%)		4/11 (36%)		20/32 (63%)		0.2
**Anti-Rejection (IS)** n (%)						
Ciclosporin		13/16 (81%)				
Tacrolimus		2/16 (12.5%)		56/59 (95%)		<0.0001
Everolimus		1/15 (7%)		10/53 (17%)		0.4
Mycophenolate mofetil		9/15 (67%)		40/53 (75.5%)		0.3
Methotrexate		3/15 (20%)				
Azathioprine		1/15 (7%)		4/53 (7.5%)		1
Corticosteroids		11/11 (100%)		43/44 (98%)		1
Everolimus				1/44 (2%)		
**Time to PTLD, Month** Q2 (Q1–Q3)						
HSCs		4 (3–5.3)				
Lung		19 (2–134)				
Kidney		98 (37–190)				
**Histology** n (%)						
**Morphology**						
IM-like		1/17 (5.9%)				
DLBCL		13/17 (76%)				
Burkitt Lymphoma		1/17 (5.9%)				
T Lymphoma		1/17 (5.9%)				
Hodgkin disease		1/17 (5.9%)				
EBV status in tumor						
EBV +		6/10 (60%)				
EBV −		4/10 (40%)				
**PTLD Treatment**						
Rituximab		5/12 (42%)				
Rituximab + Chemotherapy		5/12 (42%)				
Chemotherapy		2/12 (16%)				

**Table 2 pathogens-11-00928-t002:** Characteristics of the 17 selected case patients. A patient is considered positive (+) for sZEBRA (soluble ZEBRA) if the ELISA OD_450_ on a serum sample is greater than 0.211 (mean + 2SD of 30 seronegative plasmas, corresponding to 20 ng/mL). The measurement was performed on the sample closest to the PTLD. EBV D/R—EBV status of donors/recipients; EBV DNAemia—viral load measured by PCR (positivity threshold: 200 copies/mL, corresponding to limit of detection); DLBCL—diffuse large B-cell lymphoma; IM-like—infectious mononucleosis-like; d—days; m—months; y—years; E—early; L—late; HSCs—hematopoietic stem cells; T Ly—T-cell lymphoma, Unknown EBV tumor or serological status (?); ND—not determined.

Patient	Gender/Age	Organ	Histology/EBV Status	EBV D/R	Time to PTLD	sZEBRA	EBV DNAemia
GACY1004	21	Lung	DLBCL/?	+/−	2 m (E)	+	−
PTLD 2	F/62	HSCs	DLBCL/+	+/+	4 m (E)	+	+
PTLD 3	M/21	Lung	DLBCL/+	?/+	1.5 y (L)	−	−
PTLD 4	F/70	Lung	DLBCL/+	?/+	11 y (L)	+	+
PTLD 5	M/19	Kidney	DLBCL/+	−/+	8 y (L)	+	+
PTLD 6	F/56	HSCs	IM-like	+/+	3 m (E)	+	+
PTLD 7	F/47	Kidney	DLBCL/+	+/−	3 y (L)	+	+
PR1	M/57	Kidney	T Ly/−		16 y (L)	+	−
MS2	F/69	Kidney	DLBCL/−	+/+	6 m (E)	+	−
PV3	F/64	Kidney	Burkitt/−	+/+	11 y (L)	+	−
ZA4	M/53	Kidney	Hodgkin/+	−/+	19 y (L)	+	+
CA5	M58	Kidney	DLBCL/−	+/+	8 y (L)	+	−
R5934	M/75	HSCs	DLBCL/+	?/+	1 m(E)	−	ND
R3338	M/34	HSCs	DLBCL/+	?/+	6 m (E)	+	ND
R1192	M/18	HSCs	DLBCL/+	?/+	4 m (E)	+	ND
R2530	F/64	HSCs	DLBCL/+	?/+	4.5 m (E)	+	ND
R1767	F/65	HSCs	DLBCL/+	?/+	5 m (E)	−	ND

**Table 3 pathogens-11-00928-t003:** Comparison of EBV markers between cases and controls. The threshold of positivity of the sZEBRA (soluble ZEBRA) test was defined by the average OD_450_ obtained from 30 EBV-negative sera present in each ELISA experiment (0.115), plus 2 standard deviations (0.048), which corresponded to an OD_450_ = 0.211. The mean OD_450_ of sZEBRA was significantly higher in patients with PTLD than in controls (*p* < 0.0001), while EBV viral load in whole blood was not different between these 2 groups. PCR—viral DNA load; IQR—interquartile range; SD—standard deviation.

EBV Markers		Cases	Controls	*p*-Value
Positive whole blood EBV DNA load n (%)		6/12 (50.0%)	42/91 (46.2%)	0.22
EBV DNA in whole blood, UI/mL	Median [IQR]	0 [0, 1500]	0 [0, 1246]	
	Mean (+/− SD)	1000 (+/− 1000)	3509 (+/− 1241)	0.26
Detectable sZEBRA antigen n (%)		14/17 (82.4%)	5/91 (5.5%)	<0.0001
sZEBRA, OD_450_	Median [IQR]	0.28 [0.24, 0.34]	0.12 [0.09, 0.15]	
	Mean (+/− SD)	0.27 (+/− 0.02)	0.13 (+/− 0.01)	<0.0001

**Table 4 pathogens-11-00928-t004:** Summary table of diagnostic values, according to the selected positivity threshold and relative prevalence. Predictive values were calculated for the prevalence of PTLD in the study population (16%), or for the prevalence in the general population of transplant patients (1%). PPV and NPV—positive and negative predictive values; SD—standard deviation; CI—confidence interval.

Threshold	Sensitivity %	Specificity %	Relative Prevalence %	PPV %	NPV %
**OD = 0.211** (Mean + 2SD)	82.35 (95% CI [58.97, 93.81])	94.51 (95% CI [87.78, 97.63])	1	14.2	99.8
			16	74	97
**OD = 0.259** (Mean + 3SD)	64.71 (95% CI [41.30, 82.69])	97.80 (95% CI [92.34, 99.61])	1	24.8	99.6
			16	85	94

**Table 5 pathogens-11-00928-t005:** Case and control patients included in the study. The sZEBRA antigen was measured in the plasma of 17 transplant patients with PTLD (the samples were collected a median of 18 days before PTLD (IQR: 2–56)) and 91 transplant patients without PTLD. A—Nantes University Hospital; B— Strasbourg University Hospital; C—French CRYOSTEM Consortium; D—UMC Utrecht; E— Grenoble University Hospital; HSCs—hematopoietic stem cells. Control samples (center E) were collected prospectively at the University Hospital of Grenoble.

Center	Transplantation	Number of Samples Cases
A	Pulmonary	1
B	Renal	5
C	HSCs	5
D	Pulmonary, renal, HSCs	6 (2, 2, 2)
**TOTAL:**		17
E		**Number of Samples Controls**
	HSCs	38
	Renal	20
	Pulmonary	33
	**TOTAL:**	91

## Data Availability

Not applicable.

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
