# Peer review of "High Predictive Value of the Soluble ZEBRA Antigen (Epstein-Barr Virus Trans-Activator Zta) in Transplant Patients with PTLD"

_pathogens, 2022, doi:10.3390/pathogens11080928_

Round 1

Reviewer 1 Report

In introduction line 79, the authors point that DNA based techniques “lacks sensitivity in EBV-negative lymphoma”. It is not clear how monitoring EBV protein ZEBRA, overcomes this limitation.

The authors use the term sZEBRA, but no explanation or abbreviation is there until readers reach the discussion part. The term should be explained in the abstract and results instead.

Regarding the EBV DNA in the examined samples, the authors put the positivity threshold at 200 copies/ml, corresponding to limit of detection. The problem is that in many patients with EBV associated malignancies EBV DNA is around 100 copies/ml. Therefore, two options can be taken. The first is to improve their level of detection for EBV DNA so a more valid conclusion can be drawn. The second option is that in the discussion the authors will add a sentence explaining that their inability to detect EBV DNA below 200 copies/ml might be the cause for negative EBV DNA in some of the samples.  

Looking at the methods, the authors write “The limit of detection was 200 copies/ml, and the limit of quantification was 500 copies/ml.” so who really the tables were generated? This should be written in the legends.

In figure 1, it seems that many controls have sZEBRA around 0.2. the authors should add a paragraph in the discussion part, where they should hypothesis this phenomenon. In addition, they should try to explain why some PTLD patients have very low sZEBRA.

Author Response

Q1In introduction line 79, the authors point that DNA based techniques “lacks sensitivity in EBV-negative lymphoma”. It is not clear how monitoring EBV protein ZEBRA, overcomes this limitation.

R1 In our first paper (Habib et al. 2017), we showed that most of patients having EBV-neg PTLD were sZEBRA-positive (and EBV PCR neg). We observed the same tendency in this investigation (see patients  PR1, MS2, PV3 and CA5 in table 2). Our hypothesis is the existence of lytic EBV in the vicinity of the tumor and the lymph nodes at the periphery.

Q2The authors use the term sZEBRA, but no explanation or abbreviation is there until readers reach the discussion part. The term should be explained in the abstract and results instead.

R2 sZEBRA was defined in the abstract (lines 23 & 24)

Q3 Regarding the EBV DNA in the examined samples, the authors put the positivity threshold at 200 copies/ml, corresponding to limit of detection. The problem is that in many patients with EBV associated malignancies EBV DNA is around 100 copies/ml. Therefore, two options can be taken. The first is to improve their level of detection for EBV DNA so a more valid conclusion can be drawn. The second option is that in the discussion the authors will add a sentence explaining that their inability to detect EBV DNA below 200 copies/ml might be the cause for negative EBV DNA in some of the samples. Looking at the methods, the authors write “The limit of detection was 200 copies/ml, and the limit of quantification was 500 copies/ml.” so who really the tables were generated? This should be written in the legends.

R3 We wrote in the legend: "EBV DNAemia: viral load 113 measured by PCR (positivity threshold: 200 copies/ml, corresponding to limit of detection),

Q4 In figure 1, it seems that many controls have sZEBRA around 0.2. the authors should add a paragraph in the discussion part, where they should hypothesis this phenomenon. In addition, they should try to explain why some PTLD patients have very low sZEBRA.

R4 We modified the figure 1 to make that clearer (threshold at 0.211)

Reviewer 2 Report

Summary:

The authors present a descriptive study regarding the correlation between PTLD and the presence of EBV ZTA protein in the plasma of transplant patients. The analysis of 17 PTLD patients against 91 controls highlighted the specificity of soluble ZTA protein in the patient’s plasma as a predictive biomarker for PTLD. Early diagnostic of PTLD is critical and remains an important focus for better patient care and the findings in the manuscript are therefore potentially important. However, there are few caveats to the study that need to be addressed.

Major comments:

  • Table 1: Out of the 17 selected PTLD patients, it is my understanding that a lot of characteristics are missing. For instance, for the Risk factor “mismatch EBV D/R” only 9 out of 17 patient have the information known. Similarly, status of EBV-positivity in the tumor is only marked for 10 out of 17 patients. I understand that there is some limitations regarding the availability of the data for retrospective cohort, but could the authors address this by adding a line in the cohort description in the materials and Methods section.

  • Can the authors provide the standard curves generated for Figure 1 for the direct correlation between OD(450nm) and concentration of soluble ZTA in their ELISA. In their previous study (https://doi.org/10.1038/s41598-017-09798-7), the authors determined the threshold for the same assay to be 40ng/mL instead of 20ng/mL. Do the differences remain statistically significant and specific with a 40ng/mL threshold? If the goal is to generate a new standard of detection, the assay needs to remain robust.

  • The test is based on soluble ZTA protein. As introduced by the authors, ZTA is the viral transcription factor that initiate EBV lytic replication. Where does that protein found in the plasma come from? Do the authors claim the protein is secreted during lytic replication? OR is it just due to the release of intracellular material upon cell death during lytic replication. If it’s the latter, authors should investigate other protein candidates, such as RTA (another Immediate Early EBV transcription factor), to really convince us that ZTA is the good/best target. In both cases, the statement needs solid references.

  • Discussion, lines 184-186: The authors mentioned their recent work showed that ZTA could be excreted. This needs a clear reference of the work. How is ZTA excreted from infected cells?

  • The authors did not find a correlation between EBV viral load and PTLD occurrence (Lines 202-206) in their cohort. It goes against multiple previously published studies. Authors should try to address and explain this discrepancy in the discussion.

Minor comments:

  • Line29: “PPV” acronym should be defined here with the first use.
  • Line 41: “Lifestyles” should be “phases”.
  • Line 57: Authors should add a couple of sentence of introduction on PTLD to shortly present the general prognosis, such as lines164-167.
  • Line 160: “for” should be “in”.

Author Response

Q1 Table 1: Out of the 17 selected PTLD patients, it is my understanding that a lot of characteristics are missing. For instance, for the Risk factor “mismatch EBV D/R” only 9 out of 17 patient have the information known. Similarly, status of EBV-positivity in the tumor is only marked for 10 out of 17 patients. I understand that there is some limitations regarding the availability of the data for retrospective cohort, but could the authors address this by adding a line in the cohort description in the materials and Methods section.

R1 Table 1 has been modified accordingly

Q2 Can the authors provide the standard curves generated for Figure 1 for the direct correlation between OD(450nm) and concentration of soluble ZTA in their ELISA. In their previous study (https://doi.org/10.1038/s41598-017-09798-7), the authors determined the threshold for the same assay to be 40ng/mL instead of 20ng/mL. Do the differences remain statistically significant and specific with a 40ng/mL threshold? If the goal is to generate a new standard of detection, the assay needs to remain robust.

R2 Good point. We added this calibration curve in the supplementary materials. We increased the sensivity with the new ELISA format (threshold at 20 ng/mL)

Q3 The test is based on soluble ZTA protein. As introduced by the authors, ZTA is the viral transcription factor that initiate EBV lytic replication. Where does that protein found in the plasma come from? Do the authors claim the protein is secreted during lytic replication? OR is it just due to the release of intracellular material upon cell death during lytic replication. If it’s the latter, authors should investigate other protein candidates, such as RTA (another Immediate Early EBV transcription factor), to really convince us that ZTA is the good/best target. In both cases, the statement needs solid references.

R3 We added a sentence in the discussion (lines 203-307)

Q4 Discussion, lines 184-186: The authors mentioned their recent work showed that ZTA could be excreted. This needs a clear reference of the work. How is ZTA excreted from infected cells?

R4 We suppressed this sentence, as we have no clear evidence in vivo. We investigated this phenomenon in vitro with Akata cell lines by measuring sZEBRA 6h after stimulation in the supernatant (without evidence of cell apoptosis). Data not shown

Q5 The authors did not find a correlation between EBV viral load and PTLD occurrence (Lines 202-206) in their cohort. It goes against multiple previously published studies. Authors should try to address and explain this discrepancy in the discussion.

R5 It was discussed in our first paper (Sci Report 2017) and we put sentences in the discussion

See lines 237-240: "In this last study, viral load in whole blood was not associated with the presence of 237 sZEBRA in plasma. This could be due to the different matrix used for these two analyses 238 (PCR and ZEBRA), as we do not find a correlation between the viral load in the cell 239 compartment and the plasma, as already described [45].

Minor comments:

  • Line29: “PPV” acronym should be defined here with the first use. OK corrected
  • Line 41: “Lifestyles” should be “phases”. OK corrected
  • Line 57: Authors should add a couple of sentence of introduction on PTLD to shortly present the general prognosis, such as lines164-167. OK added  see lines 58 & 59 "Diagnosis and treatment of PTLD remains difficult with a mortality rate close to 30% [10,11]. EBV is present in approximately 80% of 59 these PTLDs."
  • Line 160: “for” should be “in”. OK

Round 2

Reviewer 2 Report

Authors made some efforts to address my comments, but I still have an issue with one item:

Q4 Discussion, lines 184-186: The authors mentioned their recent work showed that ZTA could be excreted. This needs a clear reference of the work. How is ZTA excreted from infected cells?

R4 We suppressed this sentence, as we have no clear evidence in vivo. We investigated this phenomenon in vitro with Akata cell lines by measuring sZEBRA 6h after stimulation in the supernatant (without evidence of cell apoptosis). Data not shown

The use of data that is not presented is not acceptable. Please include that data as supplemental material for examination. Release of ZTA is an important point and need to be shown properly.

Author Response

Q4 Discussion, lines 184-186: The authors mentioned their recent work showed that ZTA could be excreted. This needs a clear reference of the work. How is ZTA excreted from infected cells?

We added results obtained in table S2 suppl. materials

We extensively corrected the english througout the manuscript.

Round 3

Reviewer 2 Report

The are discrepancies between the previous Rebuttal answer and the actual supplemental figure. In the previous answer, authors declared:

"Q4 Discussion, lines 184-186: The authors mentioned their recent work showed that ZTA could be excreted. This needs a clear reference of the work. How is ZTA excreted from infected cells?

R4 We suppressed this sentence, as we have no clear evidence in vivo. We investigated this phenomenon in vitro with Akata cell lines by measuring sZEBRA 6h after stimulation in the supernatant (without evidence of cell apoptosis). Data not shown"

Major issue:

- Rebuttal mention no signs of apoptosis, yet the data presented does not provide any evidence to support the claim. Which marker was used? What experiment/controls were used. Please show the data...

Minor issue:

- Rebuttal mentions 6hrs, however table presented show 12 and 48hrs after induction. Was the intended table inserted?

Author Response

Major issue:

  • Rebuttal mention no signs of apoptosis, yet the data presented does not provide any evidence to support the claim. Which marker was used? What experiment/controls were used. Please show the data...

  • Rebuttal mentions 6hrs, however table presented show 12 and 48hrs after induction. Was the intended table inserted?

We redid the experiments, however at 6h the amount was at the limit of detection. So we did not show this result. Apoptosis was evaluated by  annexin V-FITC detection, and was negative at 6h and 12h. Positive at 48h. 

We do not think this part add relevant informations regarding the scope of the article, in this context.